# Binding and Efficacy of Anti-Robo4 CAR-T Cells against Solid Tumors

**DOI:** 10.3390/biomedicines10061273

**Published:** 2022-05-30

**Authors:** Sachiko Hirobe, Seina Nagai, Masashi Tachibana, Naoki Okada

**Affiliations:** 1Laboratory of Clinical Pharmacology and Therapeutics, Graduate School of Pharmaceutical Sciences, Osaka University, 1-6 Yamadaoka, Suita, Osaka 565-0871, Japan; hirobe-s@phs.osaka-u.ac.jp; 2Department of Molecular Pharmaceutical Science, Graduate School of Medicine, Osaka University, 2-2 Yamadaoka, Suita, Osaka 565-0871, Japan; 3Department of Pharmacy, Osaka University Hospital, 2-15 Yamadaoka, Suita, Osaka 565-0871, Japan; 4Project for Vaccine and Immune Regulation, Graduate School of Pharmaceutical Sciences, Osaka University, 1-6 Yamadaoka, Suita, Osaka 565-0871, Japan; seeeei1221@gmail.com (S.N.); tacci@phs.osaka-u.ac.jp (M.T.); 5Laboratory of Vaccine and Immune Regulation (BIKEN), Graduate School of Pharmaceutical Sciences, Osaka University, 1-6 Yamadaoka, Suita, Osaka 565-0871, Japan

**Keywords:** chimeric antigen receptor, CAR-T cell therapy, roundabout homolog 4, tumor vascular endothelial cell

## Abstract

Chimeric antigen receptor expression T (CAR-T) cell therapy has been shown be efficacious against relapsed/refractory B-cell malignant lymphoma and has attracted attention as an innovative cancer treatment. However, cells of solid tumors are less accessible to CAR-T cells; moreover, CAR-T function is decreased in the immunosuppressive state of the tumor microenvironment. Since most tumors induce angiogenesis, we constructed CAR-T cells targeting roundabout homolog 4 (Robo4), which is expressed at high levels in tumor vascular endothelial cells, by incorporating three anti-Robo4 single-chain variable fragments (scFv) that were identified using phage display. We found that binding affinities of the three CARs to mouse and human Robo4 reflected their scFv affinities. More importantly, when each CAR-T cell was assayed in vitro, antigen-specific cytotoxicity, cytokine-producing ability, and proliferation were correlated with binding affinity for Robo4. In vivo, all three T-cells inhibited tumor growth in a B16BL6 murine model, which also correlated with Robo4 binding affinities. However, growth inhibition of mouse Robo4-expressing tumors was observed only in the model with CAR-T cells with the lowest Robo4 affinity. Therefore, at high Robo4 expression, CAR-T in vitro and in vivo were no longer correlated, suggesting that clinical tumors will require Robo4 expression assays.

## 1. Introduction

Cancer immunotherapy, represented by immune cell therapy [1] and immune checkpoint molecular inhibition therapy [2], has attracted attention as a cancer treatment option. Among the former type, chimeric antigen receptor-expressing T (CAR-T) cell therapy uses T cells that are genetically engineered to express an artificial receptor [3]. CAR is a membrane receptor that tandemly binds antigen-recognition domains (ARDs), and includes a single-chain variable fragment (scFv) and intracellular signal transduction domain (STD) (i.e., CD3ζ chain) capable of inputting T cell activation signals via the hinge (HD) and transmembrane (TMD) domains. When bound to any target molecule via ARD, a signal is input in addition to a conformational change in intracellular STD, thereby promoting T-cell proliferation and cytokine secretion, exerting cytotoxic effects on target antigen-expressing cells. In 2017, CAR-T cells targeting CD19 were approved by the US Food and Drug Administration as a therapy for hematological cancers [4]. Currently, the development of CAR-T cell therapy as a treatment strategy for multiple cancer types is underway [5].

However, reports on CAR-T cell efficacy in solid tumors are limited, possibly since it is difficult for CAR-T cells to infiltrate solid tumors. For CAR-T cells to be administered intravascularly to directly contact and kill tumor cells, they must pass through the physical barriers of blood vessels and stroma. Even if CAR-T cells infiltrate solid tumors, most will produce an immunosuppressive microenvironment [6], thereby attenuating CAR-T cell function. Moreover, even though CAR-T cells are cytotoxic to target cells in vitro, in vivo they are susceptible to activation-induced cell death (AICD) associated with target-molecule recognition [7]. Therefore, to develop CAR-T cell therapy effective for solid tumors, CAR-T cells must be more efficiently delivered to tumor tissues while having appropriate CAR designs.

Focusing on the fact that the growth and survival of cancer cells depend on blood supply, we developed tumor angiogenesis-specific CAR-T cells impacting cancers (TACTICs) targeting vascular endothelial growth factor receptor-2 (VEGFR2), which is expressed at high levels in tumor vascular endothelial cells [8]. This therapy is efficacious in murine solid-tumor models, based on its ability to target tumor blood vessels and molecules associated with tumor blood-vessel injury. During target selection, we found the sarcoma to be promising, since VEGFR2 is expressed at high levels not only in tumor neovascular tissues, but also in the tumor cells themselves. However, it has been suggested that anti-VEGFR2 CAR-T cells may also react strongly with some normal human tissues [8]; therefore, prior to TACTIC clinical trials, it is necessary to search for and create tumor vascular injury type CAR-T cells targeting safer molecules with very low expression in normal tissues. We therefore focused on roundabout homolog 4 (Robo4) [9], which is expressed in tumor vascular endothelial cells at high levels, as a novel target molecule [10].

## 2. Materials and Methods

### 2.1. Cells

Human Plat-E cells (Cell Biolabs, San Diego, CA, USA) were cultured in Dulbecco’s modified eagle’s medium (DMEM; FUJIFILM Wako Pure Chemical, Osaka, Japan) containing 10% fetal bovine serum (FBS, Thermo Fisher Scientific, Waltham, MA, USA), 1 µg/mL puromycin (Merck, Darmstadt, Germany), and 10 µg/mL blasticidin (FUJIFILM Wako Pure Chemical, Osaka, Japan). B16BL6 cells (H-2^b^) derived from C57BL/6 mouse melanoma were a gift from Mochida Pharmaceutical Co., Ltd. (Tokyo, Japan). B16BL6 cells were cultured in eagle’s minimal essential medium (EMEM; FUJIFILM Wako Pure Chemical, Osaka, Japan) containing 7.5% FBS and antibiotic-antimycotic mixed stock solution (Nacalai Tesque, Kyoto, Japan). L1.2 cells (H-2^b/d^) derived from mouse leukemia were a gift from Dr. Takashi Nakayama (Faculty of Pharmacy, Kindai University, Osaka, Japan). Genes were transfected into L1.2 cells using retroviral vector (Rv)-mouse Robo4 (mRobo4)/Puro and human Robo4 (hRobo4)/Puro, carrying mRobo4 and hRobo4 genes, respectively. L1.2 cells were cultured in Roswell Park Memorial Institute 1640 (RPMI1640, FUJIFILM Wako Pure Chemical, Osaka, Japan) containing 10% FBS and antibiotic-antimycotic mixed stock solution. L1.2 cells-expressing mRobo4/hRobo4 were cultured in RPMI1640 containing 10% FBS, 5 µg/mL puromycin, and antibiotic-antimycotic mixed stock solution. All cells were maintained in a humidified atmosphere of 5% CO_2_ at 37 °C.

### 2.2. Mice

C57BL/6J (H-2^b^) and BALB/c (H-2^d^) mice were purchased from SLC (Hamamatsu, Japan). B6.PL-Thy1a/CyJ (B6 Thy1.1) mice were purchased from Jackson Laboratory (Bar Harbor, ME, USA). C57BL/6J and BALB/c were mated to produce F1 (H-2^b/d^) hybrids. Mice were maintained in the experimental animal facility at Osaka University.

### 2.3. CAR Constructs

The Robo4-specific CAR construct is shown diagrammatically in Figure 1. Anti-Robo4 CAR was used as the foundation for a second-generation anti-Robo4 CAR, which tandemly linked a CD28-derived HD/TMD with CD28-CD3ζ and CD28-CD3ζ-derived STDs using an anti-Robo4 scFv (cloneR-13, cloneR-14, cloneR-18) with a 15-amino-acid linker (V_L_-GGGGSGGGGSGGGGS-V_H_) obtained using phage display from a previous study [11]. The amino acid sequences are listed in Appendix A.

### 2.4. Preparation of CAR-T Cells

Murine CAR-T cells were produced as previously described [12,13]. Cells were suspended in complete RPMI1640 (cRPMI, RPMI1640 with 10% FBS, 50 µM 2-mercaptoethanol (2-ME, Merck, Darmstadt, Germany), and 10 U/mL recombinant murine IL-2 (PeproTech, Rocky Hill, NJ, USA)). The gene-transduced cells were cultured in cRPMI, supplemented with 5 μg/mL puromycin. We defined the end of the 24 h culture as the end of the gene transfer operation on the 0th day (day 0); medium was replaced on days 1, 3, and 5; the plate was replaced on days 2 and 4.

### 2.5. CAR mRNA Expression

The expression levels were examined via mRNA extraction and reverse transcription as previously described [13]. The CAR cDNA was detected using the Custom TaqMan Gene Expression Assay (Thermo Fisher Scientific, Waltham, MA, USA) for each anti-Robo4 scFv. We also analyzed the expression of mRNA encoded by the *Gapdh* gene, which was used as an endogenous control.

### 2.6. CAR Protein Expression

Cells were suspended in staining buffer (phosphate-buffered saline (PBS) containing 2% FBS, 0.05% NaN_3_) containing anti-murine CD16/CD32 antibody (Clone 93, BioLegend, San Diego, CA, USA) and incubated on ice for 15 min. Then, staining buffer containing Zombie Aqua Fixable Viability Kit (BioLegend, San Diego, CA, USA), phycoerythrin-cyanine 7 (PE-Cy7)-labelled anti-CD8α mAb (Clone 53-6.7, BioLegend, San Diego, CA, USA), Pacific Blue-labelled anti-CD4 mAb (Clone RM4-4, BioLegend, San Diego, CA, USA), allophycocyanin (APC)-labelled anti-hemagglutinin (HA) mAb (Clone GG8-1F3.3.1, Miltenyi Biotec, Bergisch Gladbach, Germany), or APC-labelled mouse IgG1 isotype control mAb (Clone MOPC-21, BioLegend, San Diego, CA, USA) was added and the mixture incubated on ice for 30 min. After centrifuging at 4 °C, 300× *g* for 5 min, the supernatant was removed, and cells were resuspended in staining buffer. In the flow cytometry (FCM) analysis, we gated live cells, lymphocytes, and CD8^+^ cells and measured the geometric mean fluorescence intensity (GMFI) of HA–Tag antibody or isotype control. The CAR expression intensity was quantified by calculating the GMFI ratio based on the following formula.

GMFI ratio = GMFI when using anti-HA–Tag antibody/GMFI when using isotype control

BD FACS Canto II (BD Biosciences, Franklin Lakes, NJ, USA) was used for FCM analysis.

### 2.7. CAR Binding Assays

Cells were suspended in staining buffer containing anti-murine CD16/CD32 antibody and incubated on ice for 15 min. To CAR-T cells, mouse or human Robo4/His-tag fusion proteins (Sino Biological, Beijing, China) were added at 0.05 to 0.8 μM. Then, a staining buffer containing Zombie Aqua Fixable Viability Kit, PE-Cy7-labelled anti-CD8 mAb, and AlexaFluor647-labelled anti-His tag mAb (clone OGHis, MBL Life Science, Tokyo, Japan) was added, and the suspension was allowed to stand on ice for 30 min. Cells were pelleted at 4 °C, 300× *g* for 5 min, resuspended in staining buffer, and then analyzed using FCM. The binding intensity (BI) per CAR expression level was calculated using the formula: 

BI = GMFI ratio for anti-His-tag/GMFI ratio for anti-HA–Tag

Lineweaver–Burk plots with the reciprocal of the antigen concentration on the horizontal axis and the reciprocal of BI on the vertical axis were created and used to estimate apparent maximum binding strength (appBI_max_) from the value of the intercept of the line fit to each plot (appBI_max_ = 1/intercept value). In addition, the apparent 50% binding concentration (appBC_50_) was calculated from the value of the slope of the line fit to each plot (slope = appBC_50_/appBI_max_).

### 2.8. CAR-T Cell Proliferation

Cultured CAR-T cells were suspended in cRPMI containing no IL-2 and seeded at 1 × 10⁵ cells/200 µL/well on a 96-well plate on which m/h Robo4 had been immobilized at 20–2000 ng/mL. After 16 h, BrdU solution was added, and cells were further cultured for 8 h. The supernatant was aspirated, and the intracellular BrdU uptake was measured using the Cell Proliferation ELISA, BrdU (Merck, Darmstadt, Germany).

### 2.9. Cytokine Production

Cultured CAR-T cells were suspended in cRPMI containing no IL-2 and seeded at 5 × 10⁵ cells/1 mL/well on a 24-well plate on which m/h Robo4 had been immobilized at 20–2000 ng/mL. After 24 h, the supernatant was collected. Cytokines in supernatant were measured using the OptiEIA™ Mouse IL-2 ELISA Set, OptiEIA™ Mouse IFN-γ ELISA Set, and OptiEIA™ Mouse TNF ELISA Set (BD Biosciences, Franklin Lakes, NJ, USA).

### 2.10. Cytotoxicity Assay

Assays were performed on CAR-T cell culture day 3. L1.2 cells were stained with Tag-it Violet Proliferation and Cell Tracking Dye (Biolegend, San Diego, CA, USA), mRobo4-expressing L1.2 cells, and hRobo4-expressing L1.2 cells with Cell Proliferation Dye eFluor 670 (Thermo Fisher Scientific, Waltham, MA, USA). These were used as target cells and cocultured for 18 h with 1 × 10^5^ cells each and CAR-T cells at a concentration suitable for the effector/target ratio of each well. After 18 h, Count Bright Absolute Counting Beads (Thermo Fisher Scientific, Waltham, MA, USA) were added to unify sample analysis volumes, 7-AAD Viability Staining Solution (BioLegend, San Diego, CA, USA) was added to stain dead cells, and analysis was performed using a flow cytometer. Each analysis was completed when 1000 Count Bright Absolute Counting Beads were detected in each sample. The ratio (R) of the number of mRobo4/hRobo4-expressing L1.2 cells to the number of living L1.2 cells was calculated for each well, and the cytotoxic activity was calculated from the following formula.

Cytotoxicity (%) = R (control well) − R (test well)/R (control well) × 100

Control wells were seeded with target cells only, whereas test wells were seeded with both target and effector cells.

### 2.11. Exhaustion-Marker Expression

CAR-T cells were suspended in cRPMI without IL-2, and mRobo4 was immobilized on 24-well plates at 20–2000 ng/mL. Cells were seeded at 5 × 10⁵/1 mL/well. Cells were harvested after 24 h and suspended in staining buffer containing anti-murine CD16/CD32 and incubated on ice for 15 min. Then, staining buffer containing Zombie Aqua Fixable Viability Kit, FITC-labelled anti-CD3 mAb (Clone 17A2, BioLegend, San Diego, CA, USA), APC-labelled anti-Tim3 mAb (Clone RMT3-23, BioLegend, San Diego, CA, USA), Pacific Blue-labelled anti-CD223 (LAG3) mAb (Clone C9B7W, BioLegend, San Diego, CA, USA), and PE-Cy7-labelled anti-CD279 (PD1) mAb (Clone RMP1-30, BioLegend, San Diego, CA, USA), was added, and the mixture was incubated on ice for 30 min. Cells were pelleted at 4 °C, 300× *g* for 5 min, resuspended in staining buffer, and analyzed using a FACS Canto II. Exhaustion marker expression was quantified for live and CD3^+^ cell fractions.

### 2.12. Antitumor Assays

C57BL/6 mice (7 weeks old, female) were intradermally injected with 3 × 10^5^ cells/50 µL of B16BL6 cells. Seven days after injection, CAR-T cells were injected into the tail vein at 5 × 10^6^ cells/500 µL. To assay antitumor activity against mRobo4-expressing L1.2 cells, 5 × 10⁵ cells/50 µL were injected intradermally into F1 mice (7 weeks old, female). Seven days after administration of mRobo4-expressing L1.2 cells, CAR-T cells were injected into the tail vein at 5 × 10^6^ cells/500 µL. Tumor volumes were calculated according to the following formula by measuring major and minor axes using a microcaliper. Mice with tumor sizes greater than 20 mm in any dimension were euthanized.

Tumor volume (mm^3^) = (tumor major axis; mm) × (tumor minor axis; mm)^2^ × 0.5236

### 2.13. Immunostaining

B16BL6 tumor-bearing mice treated with anti-Robo4 CAR-T cells were euthanized at 11 days after CAR-T cell injection and tumors were excised. Applied Medical Research Co., Ltd. (Osaka, Japan) prepared and stained frozen sections. Staining was performed using anti-murine CD31 (Clone ab182981, Abcam, Cambridge, UK). Sections were imaged using a BZ-X800 microscope (Keyence, Osaka, Japan). Staining was quantified using BZ-H4M software (Keyence, Osaka, Japan); five randomly selected fields of view were averaged.

### 2.14. In Vivo Persistence Assay

CAR-T cells were prepared from lymph nodes and spleens isolated from Thy1.1 mice. They were stained with Tag-it Violet Proliferation and Cell Tracking Dye for the mock group and Cell Proliferation Dye eFluor 670 for the CAR-T group. The mixtures were injected into the tail vein of B16BL6 tumor-bearing mice at 1.0 × 10⁷ cells/500 µL. Tumor tissue, spleen, and regional lymph nodes were collected 2 d after administration. Tumor tissue was dissociated using the gentle MACS protocol 37C_m_TDK_1 using the Tumor dissociation kit, mouse (Miltenyi Biotec, Bergisch Gladbach, Germany). Cells were harvested from each tissue, suspended in staining buffer containing anti-murine CD16/CD32 antibody, and incubated on ice for 15 min. Staining buffer containing Zombie Green Fixable Viability Kit (Biolegend, San Diego, CA, USA), PE-Cy7-labelled Anti-CD8α mAb (Clone 53-6.7, BioLegend, San Diego, CA, USA), and PE-labelled Anti-thy1.1 mAb (Clone HIS51, Thermo Fisher Scientific, Waltham, MA, USA), was added, and the mixture was allowed to stand on ice for 30 min.

Cells were pelleted at 4 °C, 300× *g* for 5 min, re suspended in staining buffer, analyzed using a flow cytometer. The ratio of CAR-T to mock fluorescence in the CD8α-positive and Thy1.1-positive fractions of the living cell fraction was calculated by the formula.

Ratio of CAR-T cells = CAR-T cells stained with eFluor 670)/mock T cells stained with Tag-it

### 2.15. Statistical Analysis

Significance was evaluated using the Welch’s *t*-test, Tukey’s test, two-way analysis of variance, or Dunnett’s *t*-test; details are provided in figure legends.

## 3. Results

### 3.1. Expression and Binding Characteristics of CARs

Three second-generation CARs (CAR1, CAR2, and CAR3) were constructed by tandem binding of a CD28/CD3ζ-derived STD to CD28-derived HD/TMD using each of the three scFvs isolated from the phage display as ARDs (Figure 1).

The three CAR mRNAs were expressed at similar levels after transfection into mouse T cells; expression levels were maintained through 6 days of culture (Figure 2A). When CAR protein expression was confirmed using FCM, all three were present on the T cell membrane, but the changes in expression over time differed (Figure 2A). CAR1 started with the highest expression level and decreased over time. CAR2 expression was similar to that of CAR1 on day 0, but was maintained until day 3 and then decreased. Of the three CARs, the expression level of CAR3 was the highest on day 0 but more rapidly decreased to the same level as CAR1. These results indicate that structural factors in the ARD (scFv) may influence the stability of plasma-membrane CAR and changes in membrane expression over time.

As shown in Figure 2B, the three CAR-T cells exhibited different binding properties to mRobo4 (top row) and hRobo4 (bottom row) 1 day after transfection. Binding affinities for mRobo4 in descending numerical order: CAR3 > CAR2 > CAR1. Binding to hRobo4 was different: CAR3 exhibited no binding, indicating that CAR2 has a higher binding affinity than CAR1. These results directly reflected the strength of the binding affinities of scFv to the Robo4 (Figure 1) used for the ARD of each CAR. We next assessed binding parameters (Figure 2C). The apparent maximum binding (appBImax), which represents binding capacity, was 2.5-fold higher for CAR2 and 3.6-fold higher for CAR3 than was CAR1 for mRobo4. However, the apparent 50% binding concentration (appBC50), which represents binding affinity, was almost the same for the three CARs. For human Robo4, CAR2 had 4-fold higher binding capacity than CAR1, but appBC50 values indicated that CAR1 had approximately 2-fold higher binding affinity.

### 3.2. Robo4-Specific CAR-T Cells Exhibit the Desired Therapeutic Properties In Vitro

Proliferation in response to antigen stimulation increased in all three CAR-T cells in a m/hRobo4 stimulation intensity-dependent manner (Figure 3A). The CAR-T cells with higher CAR binding capacities proliferated more actively in the hypostimulated range. Proliferation of each CAR-T cell reached a plateau at stimulation with 2000 ng/mL Robo4 protein—higher than when stimulated with 10 µg/mL anti-CD3 antibody, which also reached a plateau. We infer that this enhancement was caused by the CD28 sub-signal of each second-generation CAR.

Next, cytokine production evoked antigen stimulation was evaluated (Figure 3B). Robo4 stimulation, which induced cytokine secretion in all three CAR-T cells, had a higher threshold than proliferative induction and was below the detection limit when 20 ng/mL of Robo4 was immobilized. IFN-γ, TNF-α, and IL-2 secretion after mRobo4 stimulation were high, in the following order: CAR1 < CAR2 < CAR3. In particular, CAR3-T cells secreted more cytokines when stimulated with 2000 ng/mL mRobo4 than when stimulated with anti-CD3 antibody. Cytokine secretion after hRobo4 stimulation was higher in CAR2-T than in CAR1-T cells, reflecting the magnitude of the binding capacity of each CAR for Robo4.

With respect to Robo4-specific cytotoxic activity, all three CAR-T cells killed mRobo4-expressing target cells, with magnitudes in numerical order (Figure 3C). Cytotoxicity against hRobo4-expressing target cells was less for CAR1-T than for CAR2-T cells. To correct for differences in CAR expression, we calculated cytotoxic activity per CAR expression unit; these ratios correlated with binding capacity. Therefore, the results show that Robo4-specific cytotoxic activity exhibited by the three types of CAR-T cells is determined by the binding capacity of each CAR for Robo4 and the avidity (total antigen binding force) based on membrane expression levels.

To evaluate exhaustion associated with antigen stimulation, expression of the cell-surface exhaustion markers PD-1, LAG-3, and TIM-3 was determined (Figure 3D). For CAR3-T cells, the expression of all three markers were upregulated by 200 ng/mL mRobo4 stimulation. In contrast, 2000 ng/mL mRobo4 was required to induce a clear increase in expression in CAR1 and CAR2-T cells. Expression of exhaustion marker molecules in response to 2000 ng/mL mRobo4 stimulation increased in the order CAR1 < CAR2 < CAR3. This shows that high binding capacity of CAR for Robo4 not only enhances multiple antigen-specific CAR-T cell functions, but also promotes CAR-T cell exhaustion associated with antigen stimulation.

### 3.3. Robo4-Specific CAR-T Cell Function in a Murine Tumor Model

The relationship between Robo4 binding and in vivo antitumor effects was measured for the three CAR-T cells after being administered to B16BL6 tumor-bearing mice. Similar tumor volumes were observed for the CAR1 and CAR2 T-cells as for the mock T cell controls. Tumor growth was significantly attenuated only by the CAR3-T cells (Figure 4A). To verify whether this antitumor effect was caused by tumor vascular injury, numbers of CD31-positive tumor blood vessels on day 18 (11 days following CAR-T cell treatment) were measured (Figure 4B). Compared with mock T cell-treatment, the number of tumor blood vessels was only slightly decreased after CAR1- or CAR2-T cell treatment. In contrast, CAR3-T cell treatment significantly decreased the numbers of tumor blood vessels.

We next counted CAR-T cells administered to B16BL6 tumor-bearing mice in tumor tissue, spleen, and lymph nodes associated with the tumor 2 days later. All CAR-T cells were less abundant than mock controls in all tissues, and decreases in CAR-T cell number were inversely correlated with each CAR’s binding capacity (Figure 4C). This result suggests that CAR-T cells with higher binding to Robo4 are more likely to evoke tumor vascular injury, while also being more susceptible to AICD.

For the improvement and refinement of Robo4-specific CAR-T cell therapy, it is important to confirm the localization of Robo4 expression in multiple tissues to help select indicator cases and ensure safety. Therefore, preliminary assessment of Robo4 expression was conducted via immunofluorescence using a human normal/tumor tissue panel array (Appendix A). In the tumor types examined, hRobo4 expression was confirmed in tumor tissues. Meanwhile, expression of hRobo4 was rarely observed in normal tissues. These data suggest that CAR-T cell therapy targeting Robo4 may damage both tumor blood vessels and cancer cells without damaging normal tissue.

Robo4 is not expressed in cancer cells in the above-mentioned B16BL6 tumor model. Therefore, we investigated the effect of Robo4-specific CAR-T cell therapy when targeting both tumor blood vessels and cancer cells using L1.2 cells in which mRobo4 was forcibly expressed. Various CAR-T cells were administered to mRobo4-expressing L1.2 cancer-bearing mice, and the tumor volume was measured over time. Contrary to expectations, a significant tumor regression effect was observed in the CAR1-T cell-administered group, which had the lowest binding ability to mRobo4 (Figure 5). On the other hand, CAR2-T cell and CAR3-T cell administration, which was effective in the B16BL6 cancer-bearing model, was unable to suppress tumor growth. Therefore, the results clarified that the strength of CAR-T cell function in vitro is not always reflected at high Robo4 expression levels in tumor tissue.

## 4. Discussion

Robo4 was the fourth single-pass transmembrane receptor in the Robo family to be identified and is specifically expressed in vascular endothelial cells [10]. Its expression is particularly high in tissues with active angiogenesis, such as cancer tissues [9]. In addition, it is involved in the migration and proliferation of vascular endothelial cells and angiogenesis [14], but its detailed function is unclear. It has been shown that an antibody-drug conjugate is effective in cancer-bearing mice against mouse Robo4 [11]. Therefore, similarly to VEGFR2, which is expressed on vascular endothelial cells, Robo4 is a candidate target for tumor vascular injury-type CAR-T cells promoted by our group. In this study, we constructed three types of anti-Robo4 CAR vectors using different CAR ARDs. We evaluated the effects of the ARD on multiple functions and collected basic information on its efficacy in vitro/in vivo.

All three CARs were expressed on T-cell membranes, but their expression levels and expression profiles over time differed. ARD may influence CAR membrane expression patterns; previous studies have found that modifying the scFv failed to express it as a CAR on the membrane by CAR aggregation on the cell membrane [15]. In this study, although expression was observed for all three CARs; the expression of CAR3 disappeared rapidly, which may be attributed to CAR aggregation. Since the antitumor in vivo effect may be enhanced by stabilizing the membrane expression, it is considered necessary to confirm the cohesiveness of CAR in the future.

We found that the magnitude of binding of each CAR to Robo4 was directly reflected in the strength of each CAR-T cell function. We hypothesize that our use of the signal based on the CD28 STD of the second-generation CAR contributed to increased CAR-T cell proliferation and cytokine secretion, reflecting antigen stimulation. When targeting mRobo4 expressed in tumor blood vessels, the antitumor effect based on tumor vascular injury was only exhibited by CAR3-T cells, which also had the strongest binding to mRobo4. However, in the clinically important context of in vivo persistence, CAR3-T cells may destroy tumor blood vessels, but at the same time undergo AICD and disappear from the body. In fact, exhaustion associated with mRobo4 stimulation was most pronounced in CAR3-T cells. For anti-CD19 CAR-T cell therapy, administered CAR-T cells undergo AICD once, and then their effect is exerted by increasing numbers of remaining memory CAR-T cells [16]. In our system, to improve in vivo persistence, in might be necessary to add CD137 STD to CAR, co-express anti-apoptotic molecules, or establish an expanded culture method that is less likely to end in CAR-T cell exhaustion. Additionally, it is important to modulate epigenetic and transcriptional regulators in complex with CAR signal to reduce exhaustion [17].

One significant finding in this study is that Robo4 is expressed not only in vascular endothelial cells, but also in many human tumor cells in many tumor types. Previous studies have also shown that some cancer cells expressed Robo4 [18,19]. In recent years, it has been reported that the expression of Robo4 increases in hypoxic environments [20]. Since tumor microenvironments are typically hypoxic, it is possible that this is the reason why Robo4 was expressed in tumor cells. However, immunofluorescent tissue staining cannot distinguish whether Robo4 is present on the cell membrane or only in the cytoplasm. Considering that CAR-T cells can only act on proteins on the cell membrane, it will be necessary to confirm the localization of Robo4 expression in tumor-cell suspensions immediately following disaggregation.

Paradoxically, unlike our B16BL6 tumor experiments, CAR3-T cells with high binding to mRobo4 showed no antitumor effect, whereas CAR1-T cells with the lowest binding elicited dramatic tumor regression in tumor experiments using mRobo4-expressing L1.2 cells. This indicates that in vitro CAR-T does not necessarily predict in vivo efficacy, based on the expression level and density of the target molecule in tumor tissue. Past studies have reported that an antitumor effect was produced by CAR-T cells with high affinity for an antigen expressed at low levels [21]. Conversely, substitution with low affinity scFv for highly expressed antigens improves the efficacy of CAR-T cells [22]. The CAR3-T cells, which exhibited the best binding ability to mRobo4, showed high in vitro function, but were susceptible to exhaustion due to antigen stimulation; therefore, we speculated that AICD had been rapidly induced, reducing the efficacy in vivo. Therefore, when Robo4 expression density is high, the in vitro efficacy of CAR-T cells is not always directly reflected in vivo, suggesting that it will be necessary to thoroughly test anti-Robo4 CAR-T cells with different binding abilities based on the expression level and expression density of Robo4 in the target tumor tissue. For CAR2, the in vitro and in vivo tumor effects (B16BL6) tended to be similar to those of CAR1, but similar to CAR3 in in vivo T cell depletion (B16BL6). This suggests that CAR2 may be more exhausted than having an anti-tumor effect when administered in vivo. There are several unanswered questions regarding the relationship between in vitro and in vivo results of CAR2. In the future, in vitro and in vivo functional binding analyses are warranted to collect basic information for further insights into the functional tuning of CAR-T cells.

## Figures and Tables

**Figure 1 biomedicines-10-01273-f001:**
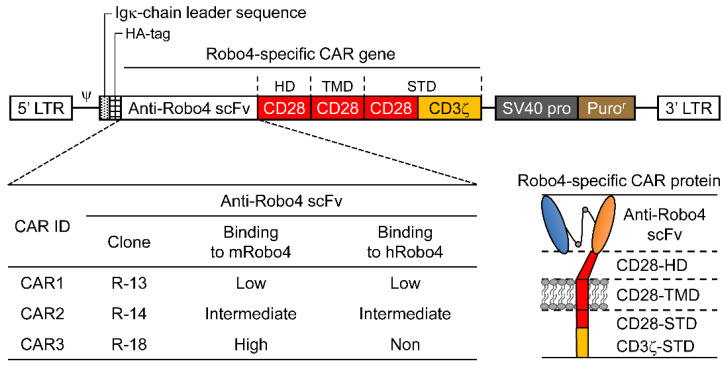
Illustration of roundabout homolog 4 (Robo4)-specific chimeric antigen receptors (CARs) with antigen-recognition domain (ARD) modification. Top, diagram of Robo4-specific second-generation CAR retroviral DNA constructs; lower left, binding of the three CAR inserts to mouse and human Robo4; lower right, diagram of Robo4-specific CAR protein. HA, hemagglutinin; HD, hinge domain; scFv, single-chain variable fragment; STD, signal transduction domain; TMD, transmembrane domain.

**Figure 2 biomedicines-10-01273-f002:**
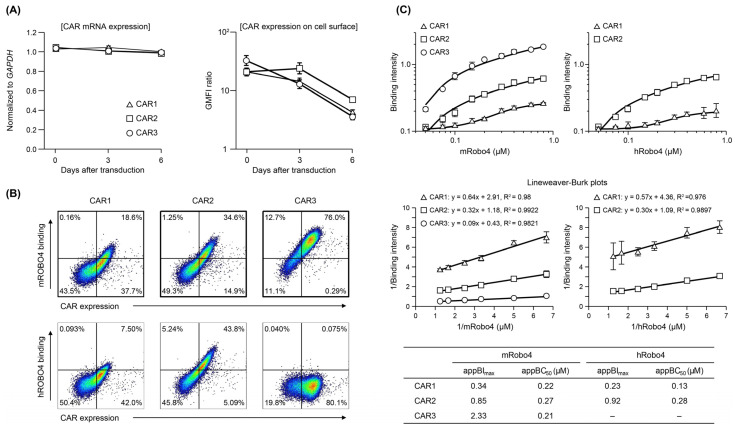
Expression and binding affinity of anti-Robo4 CARs with three different ARDs. (**A**) CAR mRNA expression analyzed using RT-qPCR, normalized to that of *Gapdh* mRNA. Cell-surface CAR measured by flow cytometry (FCM). Cells were pregated with live cells, lymphocytes, and CD8α^+^ cells. Geometric mean fluorescence intensity (GMFI) ratio was calculated as GMFI by anti-HA antibody/GMFI detected by isotype controls. (**B**) Binding to m/hRobo4 protein by anti-Robo4 CARs measured using FCM. Cells were pregated with live cells, lymphocytes, and CD8α^+^ cells. (**C**) Binding of CAR-T cells to mouse and human Robo4 measured using FCM. Binding intensity refers to the ability to bind m/hRobo4 per unit CAR expression. The apparent maximum binding strength (appBI_max_) and apparent 50% binding concentration (appBC_50_) values were calculated using Lineweaver–Burk plots.

**Figure 3 biomedicines-10-01273-f003:**
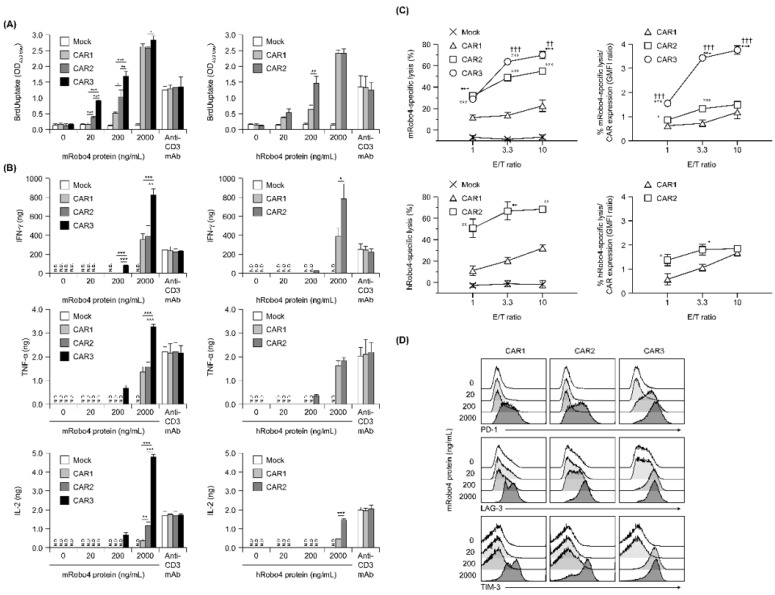
In vitro functional assays of anti-Robo4 CARs with different ARDs. (**A**) Proliferation after stimulation with mouse (left) or human (right) Robo4 protein, measured via BrdU uptake ELISA. Statistical analysis was performed using Tukey’s test for mRobo4 and Welch’s *t*-test for hRobo4; * *p* < 0.05, ** *p* < 0.01, *** *p* < 0.001. N.D., not detected. (**B**) Cytokine production following stimulation with mouse or human Robo4 protein. IFN-γ, TNF-α, or IL-2 were measured using ELISA. Left panels—stimulated with mRobo4; right panels—stimulated with hRobo4; Statistical analysis was performed using Tukey’s test and Welch’s *t*-test, respectively: * *p* < 0.05, ** *p* < 0.01, *** *p* < 0.001. (**C**) CAR-T cells 3 days after retroviral transduction were cultured with L1.2 cells and m/hRobo4-expressing L1.2 cells at the indicated E/T ratios for 18 h. Numbers of L1.2 cells and m/hRobo4^+^ L1.2 cells in the wells were measured using FCM. Specific cytotoxicity against m/hRobo4^+^ L1.2 cells was calculated from the ratio of m/hRobo4^+^ L1.2 cells to L1.2 cells. Right panels show specific cytotoxicity against CAR expression (GMFI ratios). Data are presented as mean ± SD of a triplicate. Statistical analysis was performed using Tukey’s test for mRobo4 (* *p* < 0.05, *** *p* < 0.001 versus CAR1, ^††^ *p* < 0.01, ^†††^ *p* < 0.001 versus CAR2) and Welch’s *t*-test for hRobo4 (* *p* < 0.05, ** *p* < 0.01, *** *p* < 0.001 versus CAR1). (**D**) Expression of the exhaustion markers PD-1, LAG-3, and TIM-3 on CAR-T cells measured using FCM after 24 h stimulation with mRobo4 protein at 20–2000 ng/mL. Cells were pregated live and CD3^+^ cells.

**Figure 4 biomedicines-10-01273-f004:**
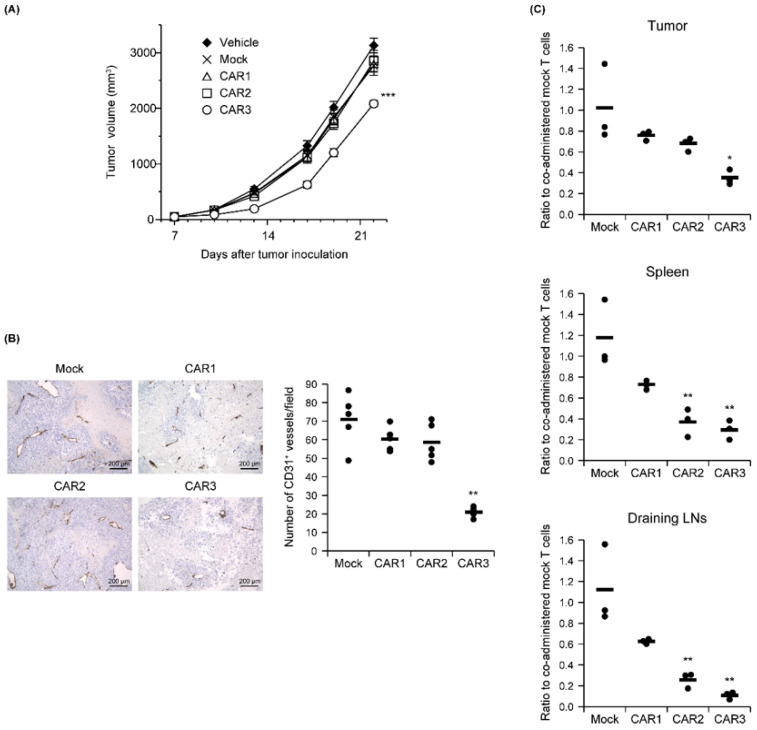
In vivo function of anti-Robo4 CARs. C57BL/6 mice bearing B16BL6 tumors were intravenously injected with anti-Robo4 CAR-T cells or control mock-T cells at 5 × 10^6^ cells/mouse on day 7 after tumor inoculation. (**A**) Tumor volumes over time. Each point represents the mean ± SE from 10 mice. Statistical analysis was performed using two-way analysis of variance: *** *p* < 0.001 vs. mock control. (**B**) Left, expression of CD31 cells in tumor tissue. Bar, 200 µm. Right, CD31 expression quantified by counting 5 fields per specimen. The circle represents an individual and the bar indicates the mean. Data represent means ± SD. Statistical analysis was performed using Dunnett’s *t*-test: ** *p* < 0.01 vs. mock group. (**C**) CAR-T cell tissue accumulations detected by FCM. Cells were pregated live, Thy1.1^+^, and CD8α^+^ cells. Ratios to co-administered mock T cells were calculated CAR-T cell treatment (cells stained with Cell Proliferation Dye eFluor 670)/mock-T cell treatment (cells stained with Tag-it Violet Proliferation and Cell Tracking Dye). The circle represents an individual and the bar indicates the mean. Statistical analysis was performed using Dunnett’s *t*-test: * *p* < 0.05, ** *p* < 0.01 vs. mock.

**Figure 5 biomedicines-10-01273-f005:**
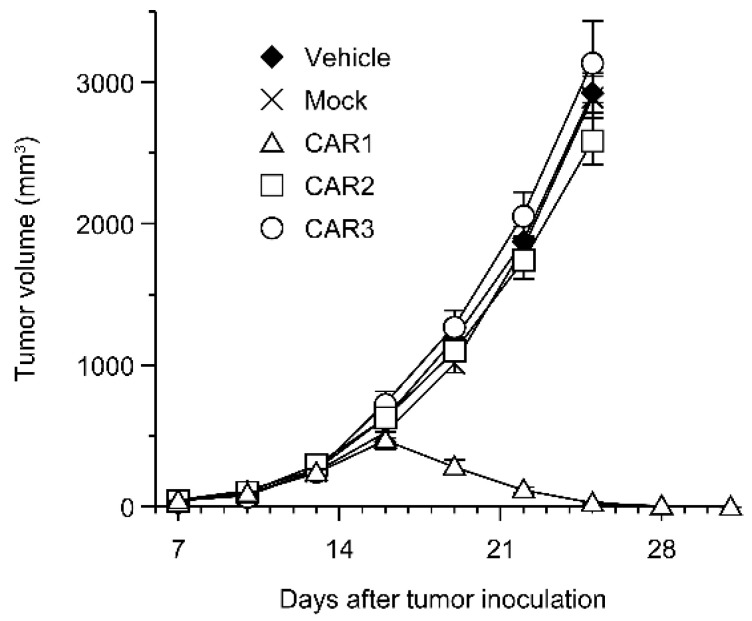
Anti-tumor effect of anti-Robo4 CARs against Robo4-expressing tumor. F1 (C57BL/6 × BALB/c) mice bearing mRobo4^+^ L1.2 tumors were intravenously injected with Robo4-specific CAR-T cells or mock-T cells at 5 × 10^6^ cells/mouse on day 7 after tumor inoculation. Each point represents the mean ± SE of 6 mice.

## Data Availability

Not applicable.

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
