# Peer review of "Binding and Efficacy of Anti-Robo4 CAR-T Cells against Solid Tumors"

_biomedicines, 2022, doi:10.3390/biomedicines10061273_

Round 1

Reviewer 1 Report

Sachiko Hirobe and co-authors present a quality and well-written experimental manuscript that describes binding and efficacy of anti-Robo4 CAR-T cells against solid tumors.

Authors constructed CAR-T cells targeting Roundabout homolog 4 (Robo4), which is expressed at high levels in tumor vascular endothelial cells, incorporating three anti-Robo4 single-chain variable fragments identified using phage display. They found that binding affinities of the three CARs to mouse and human Robo4 reflected their scFv affinities.

Authors report that when each CAR-T cell was assayed in vitro, antigen-specific cytotoxicity, cytokine-producing ability, and proliferation correlated with binding affinity for Robo4. In vivo, they found that all three T-cells inhibited tumor growth in a B16BL6 murine model, which also correlated with Robo4 binding affinities. However, inhibition of the growth of mouse Robo4-expressing tumors was present only in the CAR-T cell with the lowest Robo4 affinity.

Authors found that the magnitude of binding of each CAR to Robo4 was directly reflected in the strength of each CAR-T cell function. They hypothesized that the use of the signal based on the CD28 STD of the second-generation CAR contributed to increased CAR-T cell proliferation and cytokine secretion reflecting antigen stimulation.

Finally, authors conclude that when Robo4 expression is high, CAR-T in vitro and in vivo were no longer correlated, suggesting that clinical tumors will require Robo4 expression assays. In the future, it will be necessary to proceed with further in vitro and in vivo functional binding analyses to collect basic information that contributes to the functional tuning of CAR-T cells.

===========================

Other comments:

1) Please check for typos throughout the manuscript.

2) Authors are kindly encouraged to cite this article that describes various aspects of CAR-T cell functioning mechanisms, including with regards to treatment of solid tumors. DOI: 10.3390/cancers14041078

Overall, the manuscript is valuable for the scientific community and should be accepted for publication after edits are made.

Reviewer 2 Report

In the manuscript titled "Binding and efficacy of anti-Robo4 CAR-T cells against solid tumors", Hirobe et al present a nice set of data regarding the development of Robo4 targeted CAR-T cells for the treatment of solid malignancies. The authors propose that Robo4 is a good target for CAR-T therapy because it is highly expressed in tumour vascular endothelial cells, eliminating which would cause loss of blood flow that feeds the tumour. 

The work as presented is robust and relatively complete. There are a few places where some additional data would be helpful. Most importantly, authors make some unsupported statements based on low quality immunofluorescence data that should be revised and more fully discussed before acceptance of this manuscript. If this data is instead presented as only preliminary, the other data presented in the manuscript regarding CAR-T cells specifically is very nice and very interesting and requires only minor revision.

Specific major and minor comments: 

(1) In section 3.1 authors refer to 3 (presumably not previously reported) scFv binders against Robo4 which were raised by phage panning. Basic information on the phage panning process should be provided including some information on the recombinant proteins that was used and the phage panning workflow. Reference to previous work where similar phage panning has been used would be acceptable, but specific informtion on the protein used should be provided. 

(2) The diagram showing the Anti-Robo4 CAR design is helpful. It would be nice to also see a diagram of the Robo4 target to get an idea of its structure. Is it a single pass membrane protein? How large is its extracellular domain?

(3) In figure 1 authors report the binding for the Robo4 scFvs as low, intermediate, and high (or low/int/non for human). How was this determined? Data should be provided as supplemental figures or tables for this. ELISA binding curves for each scFv should be provided if they are available. Ideally the monomeric affinity measurements should be provided for the scFvs rather than low/int/high. 

(4) In figure 2, authors examine the binding of CAR1-3 to soluble Robo4. It would be nice to see lower concentrations on the binding curves (only seem to go down to 10 nM?) to attain the typical sigmoidal binding curve. Data is acceptable as is, but would be nicer with binding down to <0.01nM for the soluble Robo4

(5) In figure 3 authors show the response for CAR1-3 to plate bound Robo4 for cytokine production and CAR-T proliferation and cytolytic response to Robo4-expressing target cells. While this data is acceptable as is, it would be nice to also see whether proliferation and cytokine responses to Robo4 expressing cells similarly maps with scFv affinity. Given the curious results in the in vivo model (only low affinity CAR1 showing highest response to L1.4 cells in vivo), it would be informative to know whether there are in vitro differences in the responses to Robo4-expressing cellular targets. 

(6) Data presented in figure 3 is from 3 independent experiments but it is not clear what constitutes independent experiments? Were these performed with 3 different CAR-T preparations from independent donor mice? This is important to understand because some variation in CAR-T product preparation may occur due to the variables of T cell extraction, activation, and transduction. 

(7) In Figure 4 authors present data for an in vivo model wherein Robo4-CAR-T cells would be targeting tumour vascular endothelial cells, where they observe a significant drop in tumour vascularization with only the high affinity CAR3. This data is quite convincing, but would be better presented as a scatter plot rather than bar plot (Figure 4b specifically) in order to visualize the spread of the data more completely. 

(8) In figure 4c, authors present the differences in number of CAR-T cells relative to mock T cells at various sites (tumor, spleen, LN). These results are helpful but would be better presented as scatterplots showing measurements for each mouse. Were any other markers used in these experiments? Authors make some reference to CAR-T exhaustion in Figure 3 but do not address the similar question in vivo. If other markers were assessed but did not yield differences, this data should be mentioned and perhaps included as supplemental data. 

(9) In Figure 5a, authors present immunofluorescent staining of Robo4 in a normal/tumor tissue array. Images are of relatively low quality so it is difficult to assess completely. It is not clear what antibody was used here. What is "anti-Robo4-b" scFv? Is it the scFv used in CAR2 (R-14)? Authors would need to compare with a commercial anti-Robo4 antibody if they wish to support the statement that Robo4 expression is "also confirmed in domains other than vascular endothelial cells, that is, in tumor cells". 

(10) Authors claim that "hRobo4 expression was not only confirmed to be consistent with the localization of the vascular endothelial cell marker hCD31", but this is not clear. There is strong CD31 staining in some of the normal tissues, but little Robo4, whereas CD31 appears to be mostly absent in tumor tissues. Authors should revise their statements here. Also, given the quality of the images presented, this data should be presented as preliminary, though it could still be included as an assessment of Robo4 scFv binding characteristics. 

(11) In the final figure 5b of the manuscript, authors present results for a model wherein Robo4 is overexpressed in L1.2 cells. This seems quite different from 5a, and so should be separated into a different figure. What kind of cancer are L1.2 cells, I don't see it mentioned throughout the manuscript. 

(12) The results presented in Figure 5b are some of the most interesting in the paper. The observation that a lower affinity antibody may improve CAR responses in the context of a highly expressed target is very interesting. It would be important to understand why this occurred more fully. Importantly, it may be possible that this result could be peculiar to this batch of CAR-T cells, and thus it will be important to comment #6 above to confirm that there is not significant batch to batch inconsistency in the activity of CAR-T cells here. 

(13) Why is no CAR-T data provided to assess the number of CAR-T cells in the tumour or at various sites as per data in Figure 4? This could be very informative as to whether CAR1 and CAR2 cells are able to track to the tumor but become exhausted due to overactivation? 

(14) The discussion section does not adequately reference the literature on Robo4. It appears that high levels of Robo4 mRNA have been detected in some pancreatic cancer cell lines (see doi: 10.7150/ijms.28735). There appears to be other literature on the subject of whether Robo4 might be expressed in not only vascular cells but also in some tumors, and this should be more fully discussed. 

Overall, authors are commended on assembling a nice set of data, and are encouraged to address the criticisms above with consideration to feasibility. In most cases editing the language can adequately address many of the criticisms above, although some additional data may be required. 

Reviewer 3 Report

In this manuscript, the authors have developed CAR-T cells specific to tumor angiogenesis by studying the binding and efficacy of antiRobo4 CAR-T cells. Although extensive research has been done with regards to CAR-T cells, the authors' approach is unique and interesting.

Specific comments:

  1. On line 371, the authors state that "These data suggests that CAR-T cell therapy targeting Robo4 may damage both tumor blood vessels and cancer cells without damaging to normal tissue." This is an important claim and should be tested using using non-cancerous cell-line (like HeLA). In vitro tests (over 24-48 hrs) should be performed to study the effect of these CAR-T cells on non-cancerous cells.
  2. The in vivo experiment design would be easier to visualize if the authors could add a schematic in figure 4
  3. This statement needs to be corrected (Line 405), "This occurred because CAR aggregated on the cell membrane, and although expression was observed for all three CARs, the decrease in in CAR3."
  4. The in vitro data suggests that there is high exhaustion in CAR3-T cells at 200 ng/mL (Figure 3D) and 2000 ng/mL was required to induce exhaustion in CAR1 and CAR2-T cells. However, in vivo (Figure 5), there is no effect on tumor due to CAR2-T as well. The authors can elaborate more on in the discussion section.
  5. Relationship between antigen stimulation and exhaustion should be further discussed
  6. Figure 5a needs to be more quantitative and multiple areas of the same tissue should be imaged.  

Round 2

Reviewer 1 Report

Manuscript was improved and should be accepted for publication.

Reviewer 2 Report

Thank you for your revisions. This study is acceptable in present form. Hopefully follow up experiments will be fruitful for these CARs. Including the scFv sequences will also make replication of this study more feasible for other labs interested in this area. Authors are encouraged to consider sharing their Robo4 CAR plasmids through a repository such as Addgene to further encourage others to follow on this work.  

Reviewer 3 Report

The authors have satisfactorily responded to the comments.